# Something Smells Fishy: How Lipid Mediators Impact the Maternal–Fetal Interface and Neonatal Development

**DOI:** 10.3390/biomedicines11010171

**Published:** 2023-01-10

**Authors:** Maranda Thompson, Arzu Ulu, Maheswari Mukherjee, Ana G. Yuil-Valdes, Melissa Thoene, Matthew Van Ormer, Rebecca Slotkowski, Teri Mauch, Ann Anderson-Berry, Corrine K. Hanson, Tara M. Nordgren, Sathish Kumar Natarajan

**Affiliations:** 1Pediatrics Department, University of Nebraska Medical Center, Omaha, NE 68198, USA; 2Division of Biomedical Sciences, University of California Riverside, Riverside, CA 92521, USA; 3Cytotechnology Education, College of Allied Health Professions, University of Nebraska Medical Center, Omaha, NE 68198, USA; 4Department of Pathology and Microbiology, University of Nebraska Medical Center, Omaha, NE 68198, USA; 5Medical Nutrition Education, College of Public Health, University of Nebraska Medical Center, Omaha, NE 68198, USA; 6Department of Environmental and Radiological Health Science, Colorado State University, Fort Collins, CO 80525, USA; 7Department of Nutrition & Health Sciences, University of Nebraska-Lincoln, Lincoln, NE 68583, USA

**Keywords:** omega-3 fatty acid, omega-6 fatty acid, oxylipins, eicosanoids, pregnancy, fetal health, pro-resolving lipid mediator, inflammation, fish oils

## Abstract

Normal pregnancy relies on inflammation for implantation, placentation, and parturition, but uncontrolled inflammation can lead to poor maternal and infant outcomes. Maternal diet is one modifiable factor that can impact inflammation. Omega-3 and -6 fatty acids obtained through the diet are metabolized into bioactive compounds that effect inflammation. Recent evidence has shown that the downstream products of omega-3 and -6 fatty acids may influence physiology during pregnancy. In this review, the current knowledge relating to omega-3 and omega-6 metabolites during pregnancy will be summarized.

## 1. Introduction

Inflammation is necessary to defend our body against infections and other noxious insults. There are five signs of inflammation, i.e., fever, pain, redness, swelling, and loss of function. Chemokines and cytokines are mediators of inflammation that act on blood vessels, nerves, and tissue to contribute to the five classical signs [1,2]. Inflammation serves to resolve injury, remove debris, initiate tissue repair and regeneration, and suppress and prevent infection. The inflammatory response has been shown to be regulated by pro- and anti-inflammatory molecules. Previously, inflammation resolution was thought to be a passive process where pro-inflammatory molecules became diluted over time. However, recent studies have now demonstrated that anti-inflammatory molecules are synthesized at the same time and help to resolve inflammation and restore a tissue to homeostasis [3,4].

Diet is a modifiable factor in inflammation resolution. Omega-3 fatty acids (n-3 FAs) and Omega-6 fatty acids (n-6 FAs) have several important structural and functional roles in the body, and their bioactive FA metabolites are important in inflammation system regulation [5]. N-3 and n-6 FAs are incorporated into the phospholipid bilayer and released from the membrane to influence downstream pathways (Figure 1) [6]. N-3 FAs have known important roles during pregnancy, including in eye and brain development [7,8], whereas excess n-6 FAs demonstrate some detrimental effects [9,10]. The essential FAs that must be obtained from the diet are linoleic acid (LA) and alpha-linolenic acid (ALA). LA is the parent nutrient for arachidonic acid (AA) and dihomo γ-linolenic acid (DGLA). Docosahexaenoic (DHA) and eicosapentaenoic acid (EPA) are synthesized from ALA. However, ALA is not efficiently converted to EPA or DHA, with 8–10% of ALA being converted to EPA and less than 1% to DHA [11]. Therefore, consumption of preformed DHA and EPA through diet and supplementation is necessary to reach the recommended intakes [12]. Recently, evidence has linked the metabolism of n-3 and n-6 FAs into their bioactive metabolites with the inflammatory cascade [13]. These metabolites are necessary for normal physiology but can also have a role in human disease pathogenesis.

The influence of n-3 and n-6 FA metabolites during pregnancy and their role in normal and adverse outcomes is a newer topic of research. Little is known about the role of these diverse eicosanoids and how they contribute throughout gestation to maternal and infant outcomes. Therefore, understanding the interactions between controlled inflammation, the presence of n-3 and n-6 FA metabolites, and areas of dysfunction between the two in pregnancy is imperative to create useful preventative and treatment measures. The following review will detail physiological processes throughout gestation and areas of dysfunction, and the role n-3 and n-6 FA metabolites have in both. Then, to bring it all together, prematurity and pre-eclampsia will be discussed due to their relation to inflammation and the recent exploring downstream metabolite compositions in these high-risk outcomes.

## 2. The Role of FA Metabolites and Inflammation during Pregnancy

### 2.1. The Relationship between Inflammation and Nutrition during Normal Pregnancy

Normal pregnancies rely on inflammation from implantation of the blastocyst to parturition. Mor et al. categorized pregnancy into three immunological events, denoted as Stages 1–3 in Figure 2 and Figure 3 [15]. Early pregnancy is the first stage which includes the first and second trimester when implantation and placentation occur. During this stage, the body relies heavily on inflammation to allow placental cells to invade and establish the maternal–fetal blood flow. The second stage is mid-pregnancy, where rapid fetal growth and development occur within a characteristically anti-inflammatory environment. Lastly, late pregnancy is an inflammatory process as mothers begin to prepare for delivery [16,17,18].

During pregnancy, both n-3 and n-6 FAs are necessary for proper fetal development, and their role in the different stages of inflammation was described by Mor et al. For example, the n-3 FA DHA is essential for the proper development of the fetal nervous system and retina. Arachidonic acid (AA), an n-6 FA, also accumulates in the fetal brain [15]. However, an imbalance in n-3 and n-6 FA intake can influence the homeostatic equilibrium within the maternal and infant body. There are no consistent guidelines for the recommended daily intake of n-3 FAs during pregnancy, but generally, around 200 g of EPA + DHA per day is suggested [19]. In the United States, the Western diet is high in n-6 FAs and low in n-3 FAs intake [20,21]. Research has demonstrated that mothers do not reach the suggested daily intake in pregnancy [22,23], which potentially puts them at risk for poor maternal–infant outcomes due to dysregulated inflammation. Therefore, a balance between the intakes of n-3 and n-6 FAs is necessary for normal pregnancy.

Recent evidence suggests that the role of n-6 and n-3 FAs is attributable to their metabolism into bioactive metabolites involved in the inflammatory cascade [2]. The parent n-6 FAs and n-3 FAs competitively compete for the lipoxygenase (LOX), cyclo-oxygenase (COX), or cytochrome P450 enzymes (CYP450), creating eicosanoids. Omega-6 FAs generally produce pro-inflammatory eicosanoids, while n-3 FA metabolites are anti-inflammatory. The amount of n-6 to n-3 FAs influences the types of FAs that are metabolized and ultimately determines the produced eicosanoids [24]. These eicosanoids either enhance the pro-inflammatory environment or move the process toward repair and resolution. Because eicosanoids also have a role in regulating inflammation during pregnancy, more research is needed to understand how these metabolites function “normally.”

AA is a major n-6 FA that produces intermediates with a spectrum of characteristics based on the enzymatic pathway. The COX pathway produces 2-series prostanoids (i.e., prostaglandins and thromboxane). Prostaglandins have been shown to influence T-cell differentiation, platelet aggregation, and inflammatory cytokines [25]. Thromboxane A_2_ causes platelet aggregation and contributes to vascular constriction [26]. Entry into the LOX pathway creates bioactive leukotrienes and the monohydroxyeicosatetraenoic acids 5–12 (HETEs) from hydroperoxyeicosatetraenoic acid (HpETE). HETEs stimulate chemokinesis and chemotactic responses in eosinophils and neutrophils [27]. Leukotrienes are signaling molecules contributing to vascular inflammation and inflammatory diseases [28]. Lastly, the metabolism of AA through the CYP450 pathway creates hydroxy-eicosatetraenoic acids 16–20 (HETE), epoxides, and epoxyeicosatrienoic acid (EET). A well-studied HETE, 20-HETE, has demonstrated pro-inflammatory effects and influences on the vascular function [29]. EETs are further broken down by epoxide hydrolase into diols or dihyroxyeicosatrienoic acids (DHET) [30]. EETs are being studied for their vasodilation, regulation of cell differentiation, proliferation, migration, and role in cellular inflammation and apoptosis [27].

The other two major n-6 FAs are DGLA and LA. DGLA produces 1-series prostanoids in the COX pathway. Hydroxyeicosatrienoic acids (HETrE) are formed by LOX activity, while CYP-450 (CYP) produces dihydroxyeicosadienoic acid (DiHEDE). HETrE intermediates have been shown to regulate platelets and epidermal cells and the inflammatory response of neutrophils [31]. LA is the substrate for hydroxyoctadecadienoic acids (HODEs) through LOX that regulates platelet activation. Dihyroxdyoctadecenoic acids (DiHOMEs) in the CYP pathway have been found to cause mitochondrial dysfunction and promote allergic inflammation [32,33]. Figure 4 illustrates a condensed schema for the breakdown of n-6 FAs into their intermediates and bioactive compounds through the LOX, COX, and CYP enzymatic pathways.

ALA, EPA, and DHA are major substrates for intermediates and bioactive compounds (Figure 5 and Figure 6). ALA produces hydroxy-octadecatrienoic acid (HOTrE) via LOX enzymes that produce anti-inflammatory effects through the inactivation of inflammasome complexes, activate autophagy, and induce apoptosis [35]. CYP activity creates dihydroxy-octadecadienoic acid (DiHODE) whose role and tissue distribution are not completely understood [25].

EPA intermediates and resolvins mediate the effects of the immune system. LOX enzymatic breakdown of EPA yields the intermediates hydroperoxyl-eicosapentaenoic acid (HpEPE) and hydroxyeicosapentaenoic acid (HEPE) and the bioactive compounds leukotrienes (LT) A-E_5_ and lipoxins (LX) A_5_ and B_5_. The metabolite 18-HEPE is produced via CYP, and then further LOX activity reduces the adhesion of monocytes to endothelial cells, while also attenuating fibroblast activation. E-series resolvins are produced via further conversion of 18-HEPE. Resolvins E1–3 inhibit and reduce polymorphonuclear leukocyte infiltration [36,37,38].

Lastly, DHA is the substrate for the hydroxydocosahexaenoic acid (HDHA) intermediates in the LOX pathway. Of note, 17-HDHA is further metabolized into D-series resolvins by the LOX enzyme or produces protectin D1 after enzymatic hydrolysis. Resolvin D1–D6 inhibit PMN infiltration [5,39], reduce inflammatory cytokines [40,41], and influence the differentiation of immune cells [42,43]. Maresin 1 is the bioactive compound of 14-HDHA [44]. More detailed pathway information is reported in [25,34,44,45].

Specialized pro-resolving mediators (SPMs) are eicosanoids of particular interest in pregnancy due to their anti-inflammatory, repair, and resolution properties [21]. Eicosapentaenoic acid (EPA) is metabolized into the resolvin E series [RvE-], whereas docosahexaenoic acid creates maresins, protectins, and the resolvin D series [RvD-] [2]. SPMs bind to their specific G-protein-coupled receptors (GPRs) to influence downstream physiological processes [20]. SPMs act within the cell (autocrine) or locally (paracrine). In non-pregnant adults, SPM concentrations and other pathway precursors are typically reported in picogram per milliliter, with levels ranging from 20 to 2000 pg/mL in healthy adults [34,46,47]. However, the SPM concentrations are often reported as “undetectable” or below the detection limits in numerous human studies, potentially indicating their rapid consumption and transient biological presence. This difficulty in detecting SPM at biologically relevant concentrations in vivo has raised some doubt concerning the plausibility of their inflammation resolution capacity [48].

More recently, studies have begun to detail the presence of SPMs and their intermediates in the context of pregnancy. A study by Mozurkewich et al. described the differences in metabolite levels across pregnancy and demonstrated n-3 supplementation augmented 17-HDHA concentrations—a direct metabolic precursor to the D-series resolvins—but not those of other metabolites [49]. Keelan et al. investigated the effects of maternal n-3 FA supplementation on placental cytokines, SPMs, and their intermediates. DHA- and EPA-derived SPMs and their intermediates were present in the placenta, and supplementation of n-3 and n-6 FAs increased 17-HDHA and 18-HEPE relative to the controls [50]. Further, randomized clinical trials studied pre- and postnatal n-3 FA supplementation. See et al. assessed the effects of prenatal n-3 FA supplementation on metabolite levels in the offspring at birth and at 12 years of age. Prenatal n-3 FA supplementation was started at 20 weeks of gestation and continued until birth, which resulted in an increase of cord blood 18-HEPE and 17-HDHA at birth. At 12 years of age, the offspring SPM levels were not different between the offspring from supplemented mothers and those from mothers who received a placebo [51]. However, these studies did not address the influence of offspring diet on SPM levels. In another study, infants were supplemented at birth to six months (mo.) of age. The SPM levels were assessed at 6 mo., one year, and five years of age. The metabolite 18-HEPE was increased at 6 mo. in the supplemented group compared to the placebo group; however, these effects were not sustained at 1 year or 5 years of age [52]. See et al.’s study designs introduced a variety of confounders, but these studies have begun to offer insight into the presence of metabolites and the influence of supplementation. The occurrence of these SPMs during pregnancy, at birth, and six months after birth demonstrates the relevant functions of these metabolites in newborn health.

### 2.2. Implantation, Inflammation, and Fatty Acids’ Role

Using Mor et al. stages of pregnancy, implantation is part of the first stage which requires inflammation to assist in blastocyst adherence to the uterine wall and initiate processes that establish a maternal–fetal connection [17,18]. The fertilized egg, now a blastocyst, must attach to the uterine wall and achieves this through the recruitment of immune cells to the implantation site. Immune cells assist the blastocyst in getting through the epithelial lining of the uterus to the endometrium to implant [17]. In addition, immune cells also help replace the endothelium and vascular smooth muscle cells of the maternal blood vessels to establish the blood flow from the placenta to the fetus [53]. The decidua houses critical immune cells such as natural killer (NK) cells, macrophages, and dendritic cells (DCs). Polarized macrophages (Th-1) and cytokines (e.g., IL-6, IL-8, and TNF-alpha) are important inflammatory players [18]. The communication between the uterus and the blastocyst establishes pregnancy.

Evidence suggests that the different FA enzymatic pathways are key modulators of inflammation during implantation [54]. The 12/15-LOX enzymatic derivatives influence the uterine function during implantation. Li et al. reported in a mouse model an increase in 12-HETE, 15-HETE, and 13-HODE on day 4 of implantation, but the levels of 9-HODE remained low [55]. The potential mechanism is a sequence of events with the upregulation of LOX pathway enzymes by progesterone during implantation, ultimately leading to an increase in the production of eicosanoids locally, such as 12-HETE and 15-HETE. Then, the metabolites activate uterine peroxisome proliferator-activated receptors (PPARs), triggering further expression and the differentiation of cells that regulate the implantation process [54,55]. Lipoxins, synthesized from 13-HODE using the LOX pathway, are also important during implantation, as they allow vessel remodeling and placental abruption due to their ability to separate inflammation and angiogenesis [56,57]. In the 12/15 LOX knockout mice, implantation did not occur due to the necessity for spontaneous inflammation in vessel remodeling and trophoblasts anchoring [55].

The COX pathway has also been shown to influence implantation. COX-1-deficient mice are fertile but have a higher incidence of fetal malformations compared to the control. Sun et al. suggested this is due to the reduced levels of thromboxane A2 (TXA2) synthesized from the AA COX pathway because of its effects on platelet aggregation and vasoconstriction [58]. COX-2 deficiency has been shown to have contradictory results. One study found a failure of implantation and decidualization [58], whereas another study did not observe implantation failures but observed lower rates of ovulation and fertilization due to COX-2 deficiency [59]. A nonhuman primate model studied by Sun et al. suggests that endometrial COX-1 is responsible for prostaglandin (PG) synthesis, and COX-2 may impact the preparation of a receptive endometrium for implantation [58].

Omega-3 and -6 SPM metabolites undergo final metabolism through the cytochrome p450 (CYP450) pathway. Polymorphisms may participate in pregnancy and are implicated in many diseases, including cancer and cardiovascular diseases [60]. Zhou et al. described an increase in CYP2C11, CYP2C23, and CYP2J2 expression in mice on days 6, 12, and 19 of pregnancy [61]. The isoforms CYP2C8 and CYP2C9 contribute to the vasodilating epoxyeicosatrienoic acid (EETs), while CYP2C8, CYP2C9, and CYP2J2 create anti-inflammatory metabolites [60]. CYP2J2 also mediates the formation of anti-inflammatory EETs. CYP11A1 overexpression in animals or placental trophoblast has been shown to decrease trophoblast proliferation [62,63]. These data suggest a unique balance of pro-inflammatory metabolites from the different pathways is necessary for the implantation of the blastocysts.

The first trimester is a critical period of an unaffected course of pregnancy, in which implantation, trophoblast invasion, and placental development occur [64]. In the presence of dysfunctional inflammation, many complications can occur, resulting in improper implantation. Macrophages are immune cells that are either Th1 (pro-inflammatory) or Th2 (anti-inflammatory), and in pregnancy, there is a Th1/Th2 balance that is shifted heavily toward Th2 [65]. A shift towards Th1 is associated with increased levels of the pro-inflammatory cytokines interleukin 6 (IL-6), interferon-gamma (IFN-γ), interferon beta-1b (IFN-β), and TNF-alpha (TNF-α), with lower levels of anti-inflammatory IL-4 and IL-10 cytokines being associated with spontaneous abortion and preterm deliveries [66]. Oxylipins can influence placental and pregnancy pathology by transforming AA via the COX pathway to produce prostaglandins and thromboxanes related to inflammation [54].

### 2.3. Trophoblast Cells, Placental Unit, and Metabolites during Pregnancy

The proper development of the placenta is essential to establish communication and nutrient transfer between the mother and the fetus. Placental development is considered part of Mor et al.’s stage 1 of pregnancy, which relies on physiological inflammation to induce immune cell differentiation into phenotypes beneficial for trophoblasts [18]. Support for this idea comes from Fest et al., who discovered that the medium conditioned by trophoblast-induced immune cells to release pro-inflammatory cytokines essential for trophoblast development and function [67]. Trophoblast development influences placental development by modulating the maternal immune response through steps referred to as attraction, education and response [17]. First, the trophoblasts help to attract immune cells to the implantation site through the secretion of cytokines such as IL-4, IL-6, and IL-10 [68]. Next, the trophoblasts educate cells through the release of regulatory cytokines such as IL-8 and TNF-α to modulate the differentiation of immune cells [69]. Lastly, educated immune cells can respond to signals from the local environment. Fetal-derived extravillous trophoblasts invade the maternal endometrium and remain in contact with maternal immune effector cells in the decidua, allowing for communication between the mother and the fetus [69]. Normal fetal development relies on the proper development of the placenta because of its crucial role in regulating communication between the mother and the fetus.

The placenta is a modulator and can influence both the maternal and the fetal immune systems. The response of the mother and fetus is mediated by the type of response initiated by the placenta. For example, strong inflammatory responses initiated by placental infections lead to increased levels of pro-inflammatory cytokines (TNF-α, INF-γ, IL-12, and IL-6), causing placenta damage, spontaneous abortion, and preterm labor [66]. Trophoblast cell invasion and fusion are hindered by TNF-α and demonstrate increased apoptosis, increased prostaglandin levels, and increased cortisol production [70]. However, a mild inflammatory response may not terminate the pregnancy and can lead to poor outcomes such as pre-eclampsia [66]. Strong inflammation can cause preterm labor. Abnormal inflammation can influence trophoblast cells’ ability to invade the maternal arteries, leading to poor placental perfusion and the precipitation of poor maternal–fetal outcomes [71].

Studies are beginning to describe the presence and meaning of n-3 and n-6 FA metabolites in regulating fetal and placental health. Recent studies have described the roles of some n-3 FAs (EPA and DHA) oxylipins in pregnancy and the placenta as an SPM-producing tissue [49,50,72]. A review article identified other studies that suggest n-6 metabolites produced in the placenta influence the myometrium and progesterone synthesis [54]. Studies also highlight significant concentrations of CYP enzymes in the placental tissue [73,74,75]. The mechanisms of action, production, and transport of oxylipins and SPMs through the placenta remain largely unexplored. Of note, recent work by Ulu et al. demonstrated GPR18 expression—the primary receptor for the SPM RvD2—in placental extravillous trophoblast cells and umbilical cord vascular smooth muscle cells via immunohistochemical staining. In addition, this study simultaneously demonstrated the attenuation of inflammatory processes with the introduction of RvD2 in vitro using placental cell line experiments [76].

### 2.4. Balance between Anti- and Pro-Inflammatory Mediators in the Middle of Pregnancy

A typical pregnancy relies on a balance between pro- and anti-inflammatory states, and Mor et al. defined the second trimester as a predominantly anti-inflammatory environment. With the progression of pregnancy, placental activity and the production of reactive oxygen species increase due to increased fetal demand and typical development. Sufficient anti-inflammatory mediators are necessary to maintain uterine quiescence and prevent uncontrolled inflammation [77]. However, inflammation is needed to fight infection and allow the mother to maintain the fetus. An intrauterine environment shifted towards pro-inflammation at this stage can lead to intrauterine growth restriction and altered fetal growth [78]. Nutrition during pregnancy programs metabolism and physiological responses in the fetus [77]. As reviewed before, this is the period of rapid growth where n-3 and n-6 FAs are a crucial component of fetal brain development. There are no studies describing n-3 and n-6 FA metabolites’ roles in fetal development in this stage. More studies are needed to understand the baseline levels and their role in maintenance.

### 2.5. Inflammation in the Late Stages of Pregnancy and Parturition

The late stages of pregnancy are characterized as the third and final stage of Mor et al.’s categorization, as the mother prepares for the delivery of the fetus [15]. Inflammation in the mother and placenta prepares the body by initiating uterine contraction, ripening, dilating the cervix, and causing the membrane rupture. Prostaglandins, COX pathway metabolites, are vital mediators causing these changes. PGE2 and PGF2a induce contractions, cervical maturation, and fetal membrane rupture. In cord blood, 12-HETE and 15-HETrE were related to the regular course of pregnancy [79]. Further, in the setting of prematurity, the levels of umbilical cord 11-HETE and 15-HETrE were higher compared to those at term delivery. In post-term neonates, 5-HETE levels were lower, while 11-HETE levels were lower than those in preterm and term deliveries [79]. These findings suggest that lower levels of 5-HETE delay labor, whereas higher concentrations of 11-HETE and 15-HETrE accelerate it [54]. As shown by these studies, the contribution of metabolites to inflammation is necessary. However, dysregulation begins to expose the infant to a harmful environment that may increase the risk of morbidity and mortality. The following sections will discuss inflammation and metabolites in the context of specific pregnancy outcomes.

## 3. Pregnancy Outcomes and How They Relate to Inflammation and Oxylipins

### 3.1. Preterm Labor and Delivery

Currently, approximately 10% of infants are born prematurely in the United States [80]. Many different mechanisms are known to cause preterm labor; however, chemokines and cytokines have been highlighted as central to inflammation/infection-induced preterm labor—including evidence of increased placental membrane permeability during bacterial infections such as chorioamnionitis, resulting in an increased risk of preterm membrane rupture and funisitis [81,82]. There is increasing evidence that inflammation in early development may have lifelong impacts on the offspring [83]. For example, infants exposed to intrauterine inflammation have higher rates of chronic lung disease, retinopathy of prematurity, intraventricular hemorrhage, periventricular leukomalacia, and necrotizing enterocolitis than infants born equally preterm without inflammatory exposure [84,85,86]. To reduce the morbidity and mortality that inflammation-induced prematurity causes, the underlying mechanisms must be better understood.

Inflammation can be caused by infections or other signals in the intra-uterine environment. Increases in pro-inflammatory cytokines and chemokines have been implicated in preterm labor and delivery [18], whereas the anti-inflammatory cytokines IL-10 and IL-4 are reduced [1,87]. Intrauterine inflammation can be due to microbes or sterile intra-amniotic inflammation that can occur without the presence of a pathological infection. Sterile inflammation is significantly greater for women with preterm labor and intact membranes [88], while endometrial inflammation can also be a causative factor in preterm labor [89]. This could be due to other alarm signals within the intrauterine environment, which activate intracellular pathways leading to the same cytokine mediators as an infection [90]. One possible contribution to the mechanisms leading to sterile inflammation is an imbalance in nutrition and oxylipin levels.

Due to the role of inflammation in prematurity and the link between oxylipins and inflammation, studies have started to elucidate the influence of n-6 and n-3 oxylipins on early delivery. Oxylipins derived from AA have been shown in multiple studies to be associated with prematurity [79,91]. Ramsden et al. showed that at 14 weeks of gestation, levels of the 5-LOX derivatives-5-HETE and 4-HDHA above the median led to a higher risk of spontaneous preterm birth. Two other LOX derivatives demonstrated greater odds of prematurity. Levels of 15-HETE above the median and of 9-HODE below the median were associated with preterm delivery [92]. Further, Nordgren et al. demonstrated that the maternal and cord plasma SPM levels were elevated in preterm deliveries as well as in deliveries of NICU-admitted neonates [93]. Taken together, oxylipins from n-6 and n-3 FAs are tightly related to inflammation, which can become unregulated in preterm birth.

The morbidity and mortality caused by inflammation-mediated prematurity have led to an interest in targeting inflammation by upregulating anti-inflammatory mediators, blocking pro-inflammatory cytokines, or administering anti-inflammatory units. Clinically, progesterone is used to reduce the preterm delivery incidence. However, progesterone cannot block infection-induced inflammation, and concerns remain regarding route administration, bioavailability, and its association with increased incidence of stillbirths [94]. Other drugs such as cytokine-suppressive anti-inflammatory drugs (CSAIDS), anti-inflammatory prostaglandin 15-deoxy-D (12,14), prostaglandin J2 (15dPGJ2), and antibody-based TNF-α biologics are being studied for their ability to modulate inflammatory pathways and influence prematurity [18]. A large gap remains in the effective prevention and treatment of preterm labor. In a mouse model, administration of RvE3, but not of its precursor 18-HEPE, was associated with a lower incidence of preterm birth in the presence of lipopolysaccharide (LPS) [95]. By studying the effect of n-3 and n-6 FA mediator concentrations on prematurity, this research could identify a mediator that could be targeted for use in at-risk pregnancies.

### 3.2. Pre-Eclampsia

Pre-eclampsia (PE) is a clinical diagnosis characterized by inflammation, proteinuria, thrombosis, endothelial dysfunction, and placental defects in the second trimester of pregnancy [18]. Pre-eclamptic placentas are also characterized by immature trophoblast cells and placental vascular dysfunction, further altering the inflammatory milieu in these patients [96,97]. Studies have shown that the pro-inflammatory cytokines are similar between normal and pre-eclamptic pregnancies. However, in pre-eclampsia, the pro-inflammatory cytokines are highly exaggerated, and there seems to be an inhibition of anti-inflammatory cytokines [98,99,100]. In dysfunctional inflammation, trophoblast invasion and spiral artery remodeling are insufficient, causing altered blood flow and mean arterial pressures. TNF-α expression, for example, has been shown to alter prostaglandin production, the oxidant/antioxidant balance, and adhesion molecule expression in blood vessels [70]. The subsequent cascade of insufficient perfusion, hypoxia, placental stress, and release of placental factors into the maternal circulation activate more inflammation and endothelial dysfunction. The activation of various pathways can lead to inflammatory cytokine release and, ultimately, placental dysfunction, causing pre-eclampsia. Pre-eclampsia often results in preterm delivery since the only cure for pre-eclampsia is delivering the fetus and removing the placenta.

Mainly, n-6 FA metabolites have been described in plasma and placental samples in the setting of pre-eclampsia. The following research highlights the metabolites that have been identified. HETEs are AA metabolites that may result in poor placental profusion. At 20 weeks of gestation, the maternal levels of 5-, 8-, 12-, and 15-HETE were higher in pregnancies later diagnosed with PE. In the placentas, the levels of 15-HETE and 12-HETE were increased in pre-eclamptic pregnancies relative to the controls, with trophoblast cells demonstrating the same relationships [65,101,102]. In addition, 15-HETE was also found in umbilical cord blood and increase constriction of the arteries in women with PE. In contrast, Plenty et al. found no significant increase in 15-HETE in women with PE [103]. Therefore, dysregulation of HETE as a group of metabolites may underly PE.

Two recent studies present contradictory results regarding the role of lipoxins (AA metabolites) in modulating pre-eclampsia. The anti-inflammatory n-6 oxylipin lipoxin A4 (LXA4) regulates the cytokine milieu, inhibits leukocyte chemotaxis, and limits reactive oxygen species generation. Xu et al. highlighted a decrease in plasma LXA4, its receptor N-formyl peptide receptor 2 (FPR2/ALX), and the enzyme for synthesizing LXA4 in mothers with pre-eclampsia [104]. Rats treated with LXA4 had improved PE symptoms, a decrease in LPS-induced pro-inflammatory cytokines, and an increase in anti-inflammatory cytokines. However, rats with blocked LXA4 signaling developed symptoms of pre-eclampsia. In contrast, Dong et al. found women with pre-eclampsia had significantly increased levels of LXA4, TNF-α, and IL-1β [105,106]. Together, these two studies provide seemingly different results, highlighting the need for a better understanding of the enzymatic pathways in women with PE.

## 4. Conclusions

The roles of n-3 and n-6 FA intermediates and pro-resolving mediators may influence pregnancy outcomes through the regulation of inflammation. FA metabolites play an important role in the inflammation that regulates implantation, placentation, and delivery. Yet, FA metabolites have also been found to be increased in inflammatory outcomes, including prematurity and pre-eclampsia. Additional studies are needed to further understand the interactions of these metabolites in maternal–fetal biology.

## Figures and Tables

**Figure 1 biomedicines-11-00171-f001:**
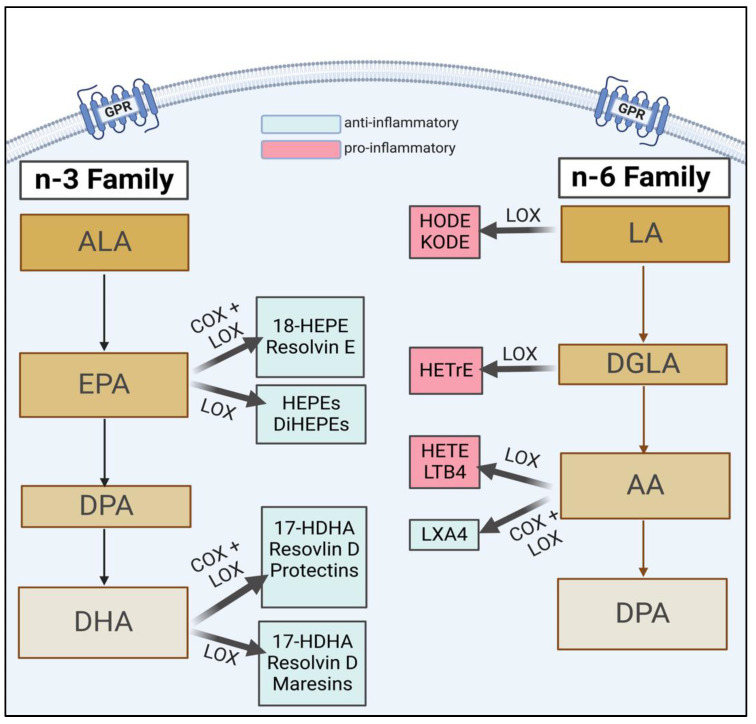
A brief schema of n-3 and n-6 lipid metabolism, highlighting key downstream FAs and their subsequent bioactive metabolites (i.e., EPA, DHA). Figure copied from Thompson et al. [14]. ALA, alpha-linolenic acid; EPA, eicosapentaenoic acid; DPA, docosapentaenoic acid; DHA, docosahexaenoic acid; HEPE, hydroxyeicosapentaenoic acid; DiHEPE, dihydroxy-eicosapentaenoic acid; HDHA, hydroxydocosahexaenoic acid; LA, linoleic acid; DGLA, dihomo-γ-linolenic acid; AA, arachidonic acid; LX, lipoxin; HODE, hydroxy-octadecadienoic acid; KODE, keto-octadecadienoic acid; HETrE, hydroxy-eicosatrienoic acids; LT, leukotriene.

**Figure 2 biomedicines-11-00171-f002:**
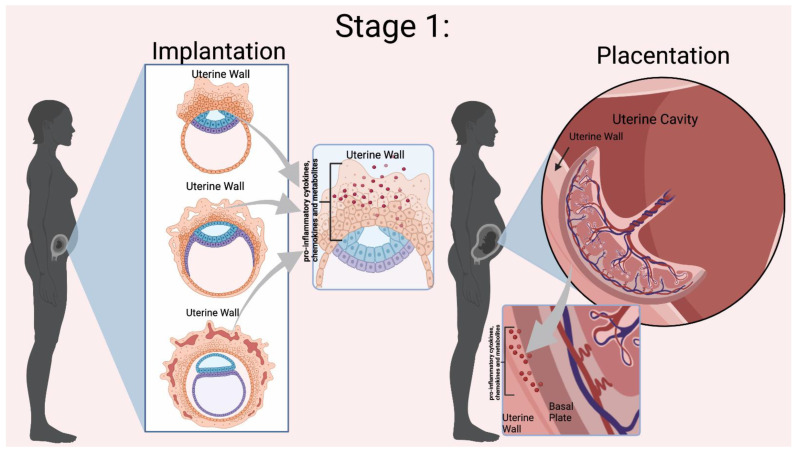
Mor et al. Stage 1 of Immunological Events Occurring During Pregnancy. Implantation relies on the presence of pro-inflammatory cytokines, chemokines, and other mediators to invade the uterine wall [10]. During placentation there is a dynamic relationship between the uterine wall and the placenta for trophoblast differentiation and proper placental development. Inflammation is indicated by the red dots in the Figure. Figure created in BioRender.

**Figure 3 biomedicines-11-00171-f003:**
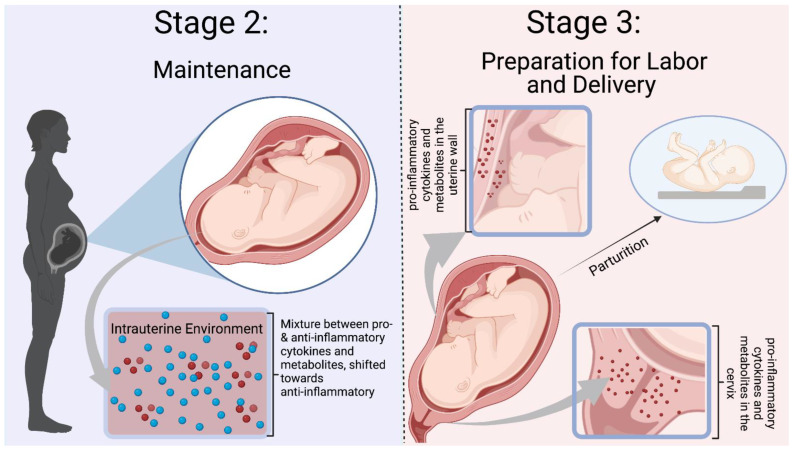
Mor et al. Stages 2–3 of Immunological Events Occurring During Pregnancy. Stage 2 is defined by a balance between pro- and anti-inflammatory mediators, with a shift towards an anti-inflammatory environment. Inflammation is needed to prevent infection and rejection of the fetus. Stage 3 is a pro-inflammatory state with preparation for labor and delivery. Cytokines and mediators remodel the cervix and infiltrate the uterus to participate in uterine contractions [10]. The blue dots represent ant-inflammatory molecules, and the red dots represent pro-inflammatory molecules. Figure was created in Biorender.

**Figure 4 biomedicines-11-00171-f004:**
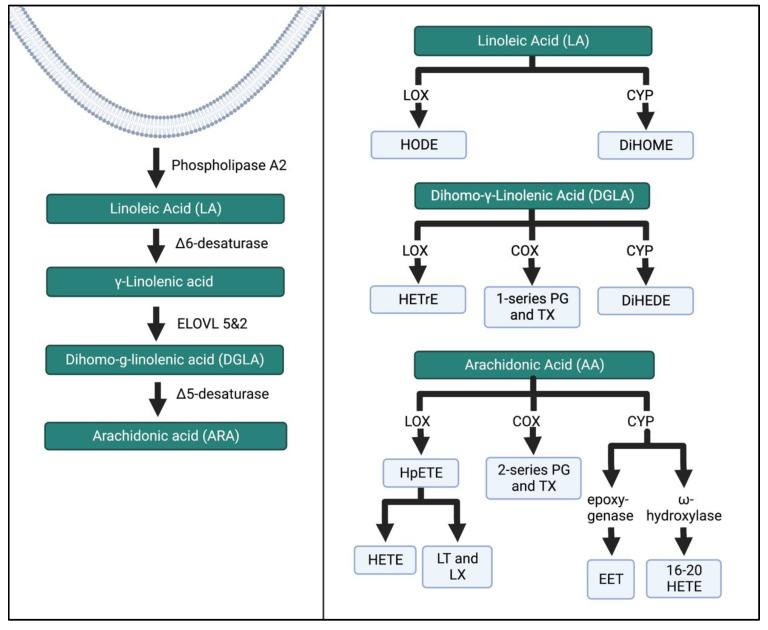
N-6 FA metabolism through the LOX, COX, and CYP enzymatic pathways. These pathways have been simplified from previous research [27,34]. ELOVL, elongation of very long chain fatty acid elongase protein; HODE, hydroxy-octadecadienoic acids; DiHOME, dihydroxy-octadecenoic acids; HETrE, hydroxy-eicosatrienoic acids; DiHEDE, dihydroxy-eicosadienoic acid; PG, prostaglandin; TX, thromboxane; HpETE, hydroperoxy-eicosatetraenoic acid; HETE, creates hydroxyeicosatetraenoic acids; LT, leukotriene; LX, lipoxin; EET, epoxyeicosatrienoic acid. The Figure was created in Biorender.

**Figure 5 biomedicines-11-00171-f005:**
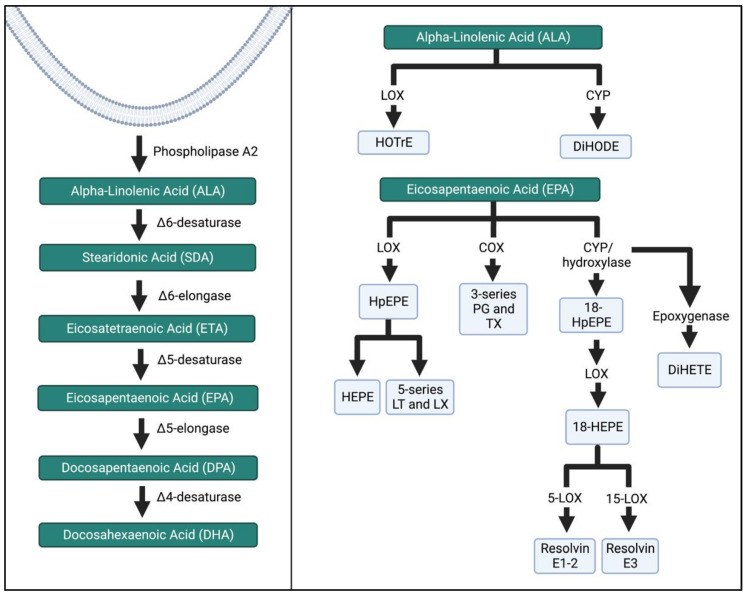
ALA and EPA metabolism by the LOX, COX, and CYP enzymatic pathways. The schema was simplified, and more detailed pathways can be found in the references listed in the Figure above. HOTrE, hydroxy-octadecatrienoic acid; DiHODE, dihydroxy-octadecadienoic acid, HpEPE, hydroperoxyl-eicosapentaenoic acid; HEPE, hydroxyeicosapentaenoic acid; LT, leukotriene; LX, lipoxin; TX, thromboxane; DiHETE, dihydroxy-eicosatetraenoic acid. The Figure was created in Biorender.

**Figure 6 biomedicines-11-00171-f006:**
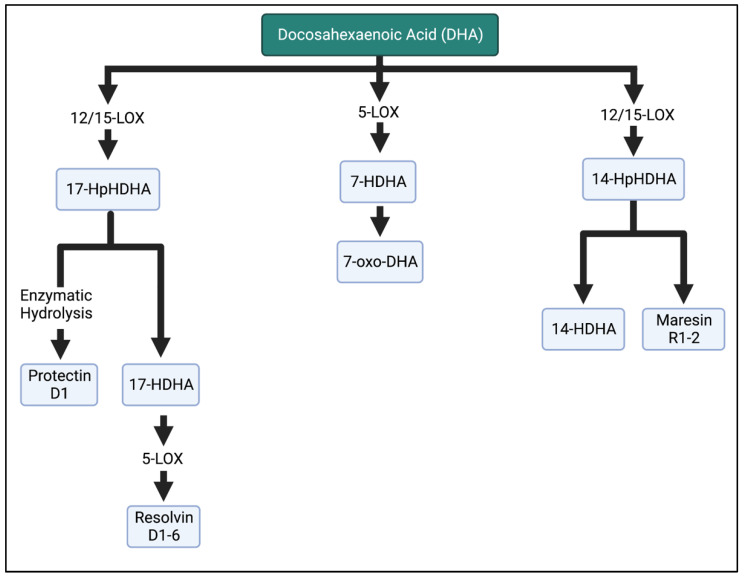
Metabolism of DHA in the LOX enzymatic pathway. DHA is metabolized into resolvins, maresins, and protectins, which are involved in the anti-inflammatory properties of n-3 FAs. HpHDHA, hydroperoxydocosahexaenoic acid; HDHA, hydroxydocosahexaenoic acid; DHA, docosahexaenoic acid. The Figure was created in BioRender.

## Data Availability

Not applicable.

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
