# Peer review of "Something Smells Fishy: How Lipid Mediators Impact the Maternal–Fetal Interface and Neonatal Development"

_biomedicines, 2023, doi:10.3390/biomedicines11010171_

Round 1

Reviewer 1 Report (Previous Reviewer 2)

None of the points I raised have been dealt with in a proper manner.

Author Response

Thank you for taking the time to review our manuscript. We feel we have addressed your comments to the extent possible, given the purpose and structure of our review, and appreciate the improvements to our work that your input facilitated.

Reviewer 2 Report (Previous Reviewer 1)

Thompson et al., “Something smells fishy: How lipid mediators impact neonatal development and the maternal-fetal interface.”

Thompson et al. in their review describe impact of omega-3 and omega-6 fatty acid on pregnancy, in either healthy and pathological state. They largely describe the role of inflammation in particular stages of pregnancy. However the impact of lipid mediator of omega-3 and omega-6 origin on pregnancy is not equally exhaustively described.

I would suggest adding scheme showing metabolic pathways for omega-6 and omega-3 fatty acids leading to creation of inflammation mediators. Figure should contains inflammatory mediators together with enzymes which catalyses their synthesis, especially these mentioned in the manuscript text.

- There is a nomenclature inconsistency. Please decide if you use FA or FAs. In lines 93,115, 116, 129, 313 you do not use abbreviations used previously in the text. Please check it through whole manuscript

- lines 104-126, proposed review aims to summarize knowledge about impact of omega-3 and omeag-6 derivatives on inflammation process during pregnancy. Therefore, this two paragraphs should contain a thorough knowledge about omega-3 and omega-6 fatty acids derivatives to allow reader better understand results of cited later in the manuscript original articles about the impact of discussed metabolites on inflammatory process and pregnancy.

Author Response

  • Thank you for the suggestion to add a metabolic scheme to the manuscript. We have inserted a scheme of this nature on line 45
  • We have revised the lines you cite to abbreviate consistently throughout the text. Use of “FA” vs “FAs” is context-dependent, based on plurality.
  • Thank you for this suggestion. We have added some background information regarding these metabolites with Figures 3-5 to help the reader with visualization and would also refer readers to the numerous references more-thoroughly discussing these extensive lipid mediators in more detail.

Reviewer 3 Report (New Reviewer)

This is a very interesting review summarizing the current knowledge relating to omega-3 and omega-6 metabolites during pregnancy.

The manuscript is clear, well illustrated and generally well written. Only minor points deserve to be improved. In particular: 

Lines 284-287: It deserves to be specified that, in bacterial infections such as chorioamnionitis, there is a weakness of placental membranes and placental barrier due to increased cytokines levels found in amniotic fluid that can lead to PROM or Funisitis (see PMID: 24768095, 26739007).

Lines 299-300: it deserves to be pointed out that also endometrial inflammation can lead to preterm delivery  (see PMID: 28466813).

3.2. Pre-eclampsia: Although authors introduced the clinical characteristcs of PE, it deserves to be added that PE is also characterised by trophoblast immaturity (PMID: 32529396) and vascular dysfunction (PMID: 34831277). This is an important point to clarify since these alterations can be due to the inflammation characterizing this pathology. 

Author Response

· Thank you for this insightful addition – we have revised this section accordingly to add that important context regarding placental permeability during periods of immune stress.

· We have similarly added reference to endometrial inflammation in lines 299-300, per your suggestion.

· These suggestions have also been added to the section in question – we thank you for your thorough review and added knowledge!

Reviewer 4 Report (New Reviewer)

In their review, Thompson and colleagues provide an overview of the role of omega-3 and omega-6 metabolites in the inflammation that regulates implantation, placentation, delivery, and abnormal outcomes during pregnancy. This review will help the field to understand how FA metabolites affect pregnancy from the very beginning until birth. I recommend publishing this important overview and have some comments below to further improve this review.

1) This review describes many FA metabolites/pathways from omega-3 and omega-6. However, we can not get an overview at the beginning of this paper. It makes it hard to reconcile all these metabolites/pathways. It should be better to set up a concept of omega-3 and omega-6 metabolism in pregnancy at the start of this paper. 

2) Although this review has almost covered all the important roles of omega-3 and omega-6 in pregnancy, non-experts readers may not immediately grasp the key points. It may help to detail some of the studies that will help illustrate their function.

Author Response

· Thank you for this comment – a similar recommendation was made by another reviewer. We have added a brief schema of n-3 and n-6 metabolism to the Introduction section.

· Thank you – we have added a brief summary regarding n-3 and n-6 function during pregnancy and various citations exploring their role in more detail.

This manuscript is a resubmission of an earlier submission. The following is a list of the peer review reports and author responses from that submission.

Round 1

Reviewer 1 Report

Thompson et al. in their review describe impact of omega-3 and omega-6 fatty acid on pregnancy, in either healthy and pathological state. They largely describe the role of inflammation in particular stages of pregnancy. However the impact of lipid mediator of omega-3 and omega-6 origin on pregnancy is not equally exhaustively described.

I would suggest adding scheme showing metabolic pathways for omega-6 and omega-3 fatty acids leading to creation of inflammation mediators. Figure should contains inflammatory mediators together with enzymes which catalyses their synthesis, especially these mentioned in the manuscript text.

I appreciate catchy titles of reviews, but sentence “something smells fishy” is not much fortunate

In few places I observed double spacing

- line 93 - It should be “n-3 FAs” not just “n-3” please check it through whole manuscript

- line 101 - I would avoid words “more recently” in the sentence “More recently, the role of n-6 and n-3 FAs have been attributed to their metabolism into bioactive metabolites involved in the inflammatory cascade”. Studies on lipid derivatives involved in inflammation has been conducted for many years.

- lines 101-119 The proposed review aims to summarize knowledge about impact of omega-3 and omeag-6 derivatives on inflammation process during pregnancy. Therefore, this two paragraphs should contain a thorough knowledge about omega-3 and omega-6 fatty acids derivatives to allow reader better understand results of cited later in the manuscript original articles about the impact of discussed metabolites on inflammatory process and pregnancy.

- lines 116-119: “In non-pregnant adults, the concentrations of SPMs and pathway precursors are reported in picogram per milliliter range, and levels have been reported as not detectable to concentrations in the low thousands in healthy adults” sentence is incomprehensible

- In lines 122-123 Authors state that supplementation with omega-3 augmented 17-HDHA without mentioning any information about this particle. Stating the character of this compound and process which it could affected, for sure will give better picture. This suggestion refers also to other results presented in the review.

- lines 137-139: “See et al.’s study designs introduces a variety of confounders, but these studies begin to offer insight into the presence of metabolites and the influence of supplementation.” please rewrite this sentence

- lines 240-242: “An anti-inflammatory environment characterizes the second stage of pregnancy defined by Mor et al., that normal pregnancy relies on a balance between the pro- and anti- inflammatory levels” sentence is incomprehensible

Some publications dedicated to the role of omega-3 and omega-6 derivatives on inflammation process during pregnancy are missing in the review, for example:

- Nordgren, Tara M., et al. "Omega-3 fatty acid supplementation, pro-resolving mediators, and clinical outcomes in maternal-infant pairs." Nutrients 11.1 (2019): 98.

- Ulu, Arzu, et al. "Omega-3 fatty acid-derived resolvin D2 regulates human placental vascular smooth muscle and extravillous trophoblast activities." International journal of molecular sciences 20.18 (2019): 4402., published by your team

Please check the newest publication in this field, I am sure that they will enriched your interesting review.

Reviewer 2 Report

Thompson et al provide a narrative review on the role of inflammation at the beginning, during and at the end of pregnancy, with a focus on bioactive metabolites of omega-3 and omega-6 fatty acids. In Introduction, the authors lay out the topic of regulation of inflammation by diet, and fatty acid metabolites in particular, in pregnancy.  The first chapter focuses on the role and regulation of inflammation in implantation and placentation, and discusses aspects of nutrition and inflammation, the role of fatty acids in implantation and inflammation, placental metabolites, and the balance between pro- and anti-inflammatory mediators in the middle of and late in pregnancy. The second chapter relates pregnancy outcomes to inflammation and oxylipins, and discussed aspects of pre-term labor and delivery and pre-eclampsia. The authors conclude “Additional studies are needed to further understand the interactions of these metabolites in maternal-fetal biology.”

Major Points

An important role in the review plays the work on specialized pro-resolving lipid mediators (SPM), as proposed by Serhan and others. Serhan’s work is referred to in refs. 2 – 9 and 22. A recent joint publication of basically everybody else in the field profoundly challenges the role of SPM’s as endogenous mediators of the resolution of inflammation (PMID 35308198). Thompson et al chose to ignore this important development in the field, and do not refer to this publication. However, a review on the field cannot be accepted without a profound discussion of the relevance of the publication mentioned.

It is well known that pro- and anti-inflammatory cytokines, not derived from the eicosanoid system play a role in pregnancy. Weighing the relative importance of eicosanoid-derived mediators to non-eicosanoid-derived cytokines would give the manuscript a novel aspect.

Throughout the manuscript, the authors accumulate data derived from all sorts of experiments or approaches: epidemiologic studies, cell experiments, experiments in rodents, intervention trials in humans, asf. The evidence thus generated is not put into a proper hierarchical system, but rather the authors seem to think that evidence generated by all sorts of approaches is equally important. Hierarchizing and weighing the current evidence, however, is the fundamental task of such a review article.